# Research Progress and Trends of Phenylethanoid Glycoside Delivery Systems

**DOI:** 10.3390/foods11050769

**Published:** 2022-03-07

**Authors:** Jin Huang, Dandan Zhao, Chaojing Cui, Jianxiong Hao, Zhentao Zhang, Limin Guo

**Affiliations:** 1College of Food Science & Biology, Hebei University of Science & Technology, Shijiazhuang 051432, China; huangjin202110@163.com (J.H.); ccjhhxx2021@163.com (C.C.); cauhjx@163.com (J.H.); 2Technical Institute of Physics and Chemistry CAS, Beijing 100190, China; zzth1@163.com; 3Institute of Agro-Production Storage and Processing, Xinjiang Academy of Agricultural Sciences, Ürümqi 830091, China

**Keywords:** drug delivery, phenylethanoid glycoside, bioavailability, biological activity, nanoemulsion, liposome, nanoparticle

## Abstract

Background: Phenylethanoid glycosides (PhGs) are obtained from a wide range of sources and show strong biological and pharmacological activities, such as antioxidant, antibacterial and neuroprotective effects. However, intestinal malabsorption and the low bioavailability of PhGs seriously affect their application. Delivery systems are an effective method to improve the bioavailability of active substances. Scope and approach: In this article, the biological activities of and delivery systems for PhGs are introduced. The application statuses of delivery systems for echinacoside, acteoside and salidroside are reviewed. Finally, the problems of the lack of uniform standards for delivery systems and the poor targeted delivery accuracy of PhGs in the current research are proposed and suggestions for future research are put forward based on those problems. Key findings and conclusions: Although there are still some problems in the delivery system of phenylethanoside, such as inconsistent standards and inaccurate delivery, phenylethanoside itself has been proven to have a variety of physiological activities. Therefore, the action mechanism and application of phenylethanoside and its delivery system should be studied further.

## 1. Introduction

Phenylethanoid glycosides (PhGs) are a class of natural glycosides containing hydroxy–methoxy-substituted phenylethyl and hydroxy–methoxy-substituted cinnamoyl groups, usually with central glucose as the parent core, and also an ester bond and oxygen glycoside bond. The molecular structure of PhGs usually comprises four parts: aglycone, central glucose, rhamnose and phenylpropyl phthalide. Natural PhG molecules contain multiple weakly acidic phenolic hydroxyl groups. These compounds are: hygroscopic; soluble in methanol, ethanol, water and other solvents with higher polarity; insoluble in organic solvents with lower polarity; easily oxidized; sensitive to strong light; and their glycosidic bonds are easily hydrolyzed under the action of enzymes [1].

At present, most of the PhGs that are isolated from nature come from worldwide folk medicinal plants, which are distributed within various families and genera, e.g., Scrophulariaceae, Rosaceae, Orobanchaceae and Plantaginaceae. In 1950, Stoll and others first isolated the phenol compound echinacoside from the roots of Echinacea. Currently, more than 500 kinds of PhGs have been isolated from nature. Typical PhGs include echinacoside, acteoside, salidroside, forsythiaside B, isoacteoside, etc. The molecular structures of several typical PhGs are shown in Figure 1.

Recent studies have shown that PhGs have a wide range of pharmacological and biological activities, such as antibacterial, anti-inflammatory, antioxidant, immunity enhancing, neuroprotective and cardioprotective effects [2]. The bioactivity of PhGs has been confirmed in vitro and in vivo. For example, forsythiaside A has shown good resistance to *Staphylococcus aureus*, *Escherichia coli*, *Pseudomonas aeruginosa*, *Micrococcus pneumoniae* and other bacteria [3]. Acteoside has been shown to effectively ameliorate neurological deficit and postpone disease onset in experimental autoimmune encephalomyelitis (EAE) mice [4]. Echinacoside has alleviated hypoxic–ischemic brain injury symptoms in neonatal rats by enhancing their antioxidant capacity and inhibiting apoptosis [5]. PhGs from *Ligustrum purpurascens* have been shown to have immunomodulatory effects on mice [6]. PhGs have also been shown to have a significant neuroprotective effect against PC12 cells in an Alzheimer’s disease (AD) model [7]. These compounds have attracted much attention because of their similar structures and strong biological activities.

Most of the active ingredients are ingested orally and need to navigate a series of barriers in the process of reaching the site of action through blood circulation. These barriers comprise: the solubility of the active ingredients; the role of acids, enzymes, microorganisms and food in the gastrointestinal tract; the absorption of gastrointestinal epithelial cells; and liver metabolism. The US Food and Drug Administration (FDA) defines bioavailability as “the speed and degree that the active ingredient or active site in a drug reaches the active site in an effective form after being absorbed”. In vivo and in vitro experiments have shown that PhGs have antioxidant activity and can protect various cells from injury or apoptosis [8]. Most of the active substances in nature can be directly added to food after being extracted from plants. Although PhGs have good biological effects, PhGs cannot exert their full therapeutic potential due to intestinal malabsorption and their low bioavailability.

Through the in-depth study of active substances and the emergence of new technologies in recent years, increasing numbers of chemical, pharmaceutical and biological methods have been developed to improve the oral bioavailability of active substances. Such delivery systems can enhance the water solubility of the active substances and avoid the influence of light, heat and oxygen. Commonly used delivery systems in food include nanoemulsions, nanoparticles, liposomes and microcapsules [9,10,11]. Embedding technology is used to insert the active substance into the micro-sized sealed wall material for the later release of its contents. Compared with chemical methods, embedding technology does not change the chemical structure of active ingredients, thus limiting changes to the physical and chemical properties of PhGs. In addition, a delivery system can effectively avoid the adverse effects of the external environment and improve the bioavailability of PhGs. Therefore, in recent years, methods for embedding active substances combined with a delivery system has been the approach favored by researchers.

The wider application of phenolic substance delivery systems has encouraged research to improve the bioavailability of PhGs. Currently, delivery systems for PhGs mainly include nanoemulsions, phospholipid complexes and solid lipid nanoparticles. These nano-sized delivery systems achieve better controlled release and slow release effects. Different delivery systems can be combined according to the characteristics of different PhGs [10]. The obtained PhG complexes are subjected to the characterization of their physical and chemical properties, in vivo and in vitro absorption and metabolism tests and the optimization of the compound prescription, with the aim of effectively improving their bioavailability.

In this article, the biological activities of and delivery systems for PhGs are introduced. The embedding technologies used for echinacoside, acteoside and salidroside, which are three landmark PhG compounds, are reviewed. Finally, the future development of PhG delivery systems is proposed.

## 2. The Biological Activities of PhGs

In recent years, many studies have shown that PhGs have antibacterial, antioxidant, neuroprotective, liver protective and memory improving effects. To aid the further study of PhGs, this section reviews the main biological activities of PhGs.

### 2.1. Antioxidant Activity

Many PhGs and extracts that are rich in PhGs have shown powerful antioxidant activity, with the ability to scavenge O^2−^, OH, H_2_O_2_ and ^1^O_2_ free radicals and protect DNA from oxidative damage caused by OH. Ahmed et al. [12] used plantain as the raw material to study the antioxidant effect of its methanol extract. The results showed that the major phenolic constituent in plantain is acteoside (ACT) and that ACT inhibited the lipopolysaccharide-induced production of nitrous oxide (NO) in RAW264.7 macrophages and scavenged both superoxide radicals and 2,2-diphenyl-1-picrylhydrazyl (DPPH) radicals. ACT has also displayed a strong scavenging effect on DPPH in vitro. Wei et al. [5] explored whether echinacoside (ECH) has a neuroprotective effect in neonatal rats. Superoxide dismutase (SOD), glutathione peroxidase (GSH-Px) and catalase (CAT) activities and malondialdehyde (MDA) production were assessed to determine the antioxidant capacity of ECH. The results showed that ECH treatment helped to recover antioxidant enzyme activities and decreased MDA production. ECH treatment could reduce the neuronal damage caused by apoptosis through its antioxidant effect. Hu et al. [13] synthesized three salidroside (SAL) derivatives using tyrosol as the starting material. Their antioxidant activities in vitro were investigated using the reducing force and DPPH free radical scavenging methods. The results showed that different concentrations of SAL and its derivatives could reduce and scavenge DPPH radicals in a concentration-dependent manner.

### 2.2. Antibacterial and Antiviral Activities

Most PhGs have antibacterial and antiviral activities and have good inhibitory effects on *S. aureus* and *Streptococcus pneumoniae*. Liu et al. [14] extracted total PhGs (TPGs) from *Monochasma savatieri Franch*. Ex Maxim for tests on their antibacterial and anti-inflammatory activity, toxicity and in vitro antibacterial activity. The results showed that TPGs had an obvious antibacterial effect on *E. coli*, *Streptococcus pneumoniae* and five other bacterial strains in the concentration range of 0.0625–16 mg/mL. TPGs could effectively inhibit *Pseudomonas aeruginosa* or *S. aureus*, thus improving the survival rate of septic mice. These results indicated that TPGs could be useful in developing drugs to treat respiratory infections or pneumonia caused by *S. aureus*. Some scholars have conducted in-depth research on the antibacterial mechanisms and drug combinations of PhGs. Fazly et al. [15] investigated the effect of lemon verbena aqueous extract, ACT and caffeine in combination with gentamicin against *S. aureus* and *E. coli* strains. They concluded that the coadministration of ACT and gentamicin had a synergistic effect against *E. coli* and *S. aureus*. Agbo et al. [16] isolated SAL from *Loranthus micranthus* Linn. and studied its antiviral activity. The results showed that SAL could inhibit a recombinant strain of the respiratory syncytial virus.

### 2.3. Neuroprotective Activity

Studies have found that PhGs can protect nerve cells in a variety of ways. Zhu et al. [17] showed that echinacoside could activate enzyme pathways in neurons, inhibit cytochrome c release caused by rotenone exposure and protect neurons and non-neurons from rotenone injury. Generally, it is believed that the β-amyloid protein is related to Alzheimer’s disease. Ju et al. [18] studied the protective effects of PhGs on damage to PC12 cells induced by aggregated β-amyloid protein 142 (Aβ(1-42)). The results showed that PhGs could significantly ameliorate cell injury, improve cell survival rate and have an obvious neuroprotective effect. Lee et al. [19] isolated and purified a *Forsythia koreana* extract to obtain Forsythia PhGs. Butein was used as a positive control, which shows DPPH radical scavenging activity and protective activity against glutamate-induced oxidative injury in HT22 cells. The results indicated that the PhGs from Forsythia had a neuroprotective effect. Li et al. [20] studied the neuroprotective mechanism of four PhGs (SAL, ECH, ACT and isoacteoside) on H_2_O_2_-induced cytotoxicity in PC12 cells. PhG pretreatment improved intracellular antioxidant enzymes and ultimately reversed H_2_O_2_-induced cytotoxicity in PC12 cells. Nuclear factor erythroid-derived 2-like 2 (NRF2) can be used as a therapeutic target in neurodegenerative diseases. PhGs have been shown to protect PC12 cells from H_2_O_2_-induced cytotoxicity by inducing NRF2 activation.

### 2.4. Hepatoprotective Activity

The liver is not only the main detoxification organ in the human body, but it also has important digestive and metabolic functions. Li et al. [21] studied the protective effects of ECH from *Cistanche deserticola* on D-galactosamine (GalN) and lipopolysaccharide (LPS)-induced acute liver injury in mice. ECH was injected intraperitoneally one hour before GalN/LPS exposure. The results showed that the GalN/LPS-induced hepatotoxicity decreased and the survival rate of the mice increased following pretreatment with ECH. Cui et al. [22] analyzed and identified the hepatoprotective effect of urinary metabolites after the oral administration of ACT to rats. Compared with native ACT, ACT metabolites exhibited higher hepatoprotective activities by regulating oxidative stress, lipid peroxidation and inflammatory responses. Eldesoky et al. [12] found that the methanol water extract of plantain, with ACT as the main active component, inhibited the serum activity of alanine aminotransferase (ALT), aspartate aminotransferase (AST) and alkaline phosphatase (ALP). The methanol water extract also attenuated CCl_4_-induced lipid peroxidation and exhibited a potent hepatoprotective effect. Many studies have confirmed that total PhGs and SAL can exert hepatoprotective activities in vitro and in vivo via antioxidation and the regulation of related protein expression [23,24,25].

PhGs have a wide range of biological activities; in addition to the above four, PhGs also have obvious anti-aging and anti-cancer effects and more. Table 1 summarizes the research statuses of different PhGs, lists the cell or mouse models used in each study and details the mechanisms of the different biological activities of the PhGs.

## 3. The Bioavailability of PhGs

Bioavailability refers to the speed and degree of the active ingredients in a drug entering the human circulation after being absorbed. It reflects the proportion of the drug that enters the human circulation. For oral drugs, it describes the percentage of the amount of the drug that is absorbed by the gastrointestinal tract and reaches the systemic circulation through the liver. In general, the systemic effect of PhGs mainly depends on their bioavailability through the gastrointestinal barrier. The absorption and metabolism of PhGs in vivo are depicted in Figure 2. The bioavailability of PhGs is extremely low because of their poor oral absorption, rapid metabolism and fast elimination from the body. The low oral bioavailability of PhGs means that it is difficult for them to reach a certain blood concentration at the site of action, thus limiting their biological activities [34]. Therefore, in recent years, researchers have explored a variety of methods to improve the bioavailability of PhGs.

### 3.1. The Absorption of PhGs

Active ingredients are absorbed into the bloodstream through the gastrointestinal tract to give full play to their curative effect. The chemical structure of PhGs determines the rate and degree of their absorption through the gastrointestinal tract. Glycosides are mostly hydrolyzed by enzymes or microorganisms in the gastrointestinal tract before being absorbed by the human body. The type, position and quantity of sugar groups also affect the absorption of PhGs. Li et al. [35] studied the absorption kinetics and oral bioavailability of ECH in rats in vivo and in vitro. The results showed that the absorption of ECH in the small intestine was poor and site-dependent and the absolute bioavailability was only 0.83% following intragastric administration. Improving the dosage form and compatibility of drugs could enhance their intestinal absorption and oral bioavailability. Fu et al. [36] used *Cistanche deserticola* polysaccharide to improve the intestinal absorption of ECH. The results showed that the plasma concentration of ECH alone was only 406.8 ng/mL; however, the plasma concentration of ECH was significantly increased after adding different polysaccharides. Huo et al. [37] studied the absorption, distribution and excretion of ACT in rats and concluded that the absorption of ACT in rats conforms to first-order kinetics. The highest concentration was detected in the intestines and lungs, followed by the stomach and muscle. A small amount was distributed in other tissues, which could be eliminated mainly through metabolic processes. Tanino et al. [38] studied the absorption of ECH and ACT in a tubular rat extract using the human intestinal Caco-2 cell monolayer method. The results showed that intact ECH and ACT were not limited by the P-gp efflux pump and the two PhGs were transported through a paracellular route. However, phloridzin dramatically reduced the absorptive permeability of intact ECH and ACT, suggesting that apical sodium/glucose cotransporter 1 (SGLT1) plays a major role in the intestinal absorption of intact ECH and ACT.

### 3.2. The Metabolism of PhGs

Oral administration is one of the most important and convenient routes of administration. However, it requires the drug to pass through the gastrointestinal tract and is affected by gastric juices, intestinal juices, intestinal flora and intestinal enzymes before entering the bloodstream. Therefore, in metabolic research, we should not only evaluate the metabolic stability of active substances in plasma, bile, urine and feces, but also evaluate the metabolic stability of the intestinal flora and intestinal enzymes to reveal the intestinal metabolic mechanisms affecting the absorption of the drug.

Metabolism in the stomach and intestinal cavity mainly refers to the process of chemical degradation or biodegradation of various components in food under the action of gastrointestinal pH-resistant enzymes. PhGs are not metabolized in stomach much as their metabolism mostly occurs in the intestinal cavity. There are two kinds of enzymes related to metabolism: enzymes secreted by the cells of the stomach, pancreas and small intestine; and enzymes provided by intestinal microflora [39]. The extensive physiological activities of PhGs have led to a series of studies on the metabolism of PhGs in vivo and in vitro. Tu et al. [40] studied both the stability of acteoside, isoacteoside and 2′-acetylacteoside in simulated gastric and intestinal juices and their metabolism by human intestinal bacteria. Eleven metabolites of acteoside, seven metabolites of isoacteoside and eleven metabolites of 2-acetylacteoside were identified, including hydroxytyrosol, hydroxytyrosol sulfate conjugation, caffeic acid, etc. There are eight possible metabolic pathways that led to the generation of these metabolites, including deglycosylation, the removal of rhamnose and the removal of hydroxytyrosol. Cui et al. [41] showed that isoacteoside (ISAT) was metabolized in the gastrointestinal tract before entering the bloodstream, which reveals the reason for its poor bioavailability. The biotransformation process of oral ISAT in vivo is as follows. Oral ISAT enters the stomach, in which there is no obvious change. It is not absorbed into intestinal blood until a series of low molecular weight compounds are formed through intestinal flora and enzyme-mediated biotransformation. These low molecular weight compounds are absorbed by intestinal epithelial cells into the hepatic portal vein, metabolized in the hepatic portal vein in phase II and then distributed throughout the body. Finally, all metabolites are excreted through urine and feces. Zhou et al. [42] analyzed the metabolic pathways of SAL when administered via gavage to rats and identified its in vivo metabolites. A total of 20 metabolites were identified. The dominant metabolic pathways of salidroside include glucuronidation, acetylation, sulfation and methylation. Thus, SAL undergoes extensive phase I and II metabolism in rats. The main excretion path is the urine, feces and bile.

Table 2 summarizes the pharmacokinetics of different PhGs in animal models from existing studies. Several studies have shown that PhGs enter the circulatory system following oral administration or injection and can be absorbed rapidly by the gastrointestinal tract, although the bioavailability is low, and then distributed to most tissues, including the brain. Therefore, the reason for the low bioavailability of PhGs is related to the limited effectiveness of intestinal absorption and the universality of metabolism.

## 4. Lipid-Based Delivery Systems

Delivery systems represent a technology that can control the distribution of active substances within an organism in a spatial, temporal and dose-related manner. It aims to deliver the right amount of active substance to the right place at the right time, thereby increasing utilization efficiency, improving curative effects and reducing toxicity and side effects. The commonly used delivery systems include emulsions (e.g., nanoemulsions, multilayer emulsions and Pickering emulsions), liposomes, nanoparticles and microcapsules. The availability of PhGs depends on the stability of their active structures and bioavailability. The poor absorption and bioavailability of PhGs in the gastrointestinal tract limit their application in production [51]. Therefore, the encapsulation of PhGs using delivery systems can increase the resistance of PhGs entering the external phase, inhibit the entry of light, oxygen and other substances into the internal phase, reduce the degradation rate of PhGs during processing, storage and transportation and control the fixed-point release in the body to improve their bioavailability [52]. Figure 3 shows three widely used delivery systems.

A nanoemulsion (particle size 50–200 nm) is a mechanically unstable system that requires external energy to break down large droplets into small droplets. Nanoemulsions have a small particle size, uniform dispersion, high dynamic stability and no obvious precipitation in the long term. However, there are still some limitations. To reduce the interfacial tension and saturate the very large interface, an emulsifier with high surface activity is usually used in the nano system and a large amount of emulsifier will inevitably have implications for food safety. Moreover, nanoparticles could change the normal absorption pathway, interfere with the absorption, distribution, metabolism and excretion of substances and cause potential uncertain changes in the organism [53]. During the preparation process, the structure of sensitive compounds may change, resulting in decreased biological activity [54].

Liposomes comprise a bilayer carrier composed of phospholipid and cholesterol. They have both lipophilic and hydrophilic cavities and can be used for the encapsulation, delivery and release of water-soluble, lipophilic and amphiphilic components. Liposomes have excellent biocompatibility and can control the release rate of entrapped active compounds [55]. Therefore, liposomes are considered as a promising candidate for a delivery system for active plant components. Liposomes have many advantages. They can embed active substances that are difficult to dissolve in water or have a bad smell into the particles and can fuse with the cell membrane to improve bioavailability, so they are widely used in functional foods, drugs, chemicals, daily necessities, etc. Despite the wide range of studies dealing with liposomes, the use of these carriers involves limitations. Their release rate is not easy to control and their encapsulation efficiency is low [56]. In addition, they are unstable in storage and they precipitate easily; therefore, their application in industrial production is limited.

Solid lipid nanoparticles (SLNs) and liquid crystalline lipid nanoparticles (LCNPs) are commonly used nanoparticles. Nanoparticles can enhance their solubility by embedding or dissolving active substances through hydrogen bonding and hydrophobic interactions. Substances can also be adsorbed or coupled with the nanoparticle surface. As a new type of drug carrier, nanoparticles not only have the advantages of drug targeting and changing the in vivo drug distribution, but they can also regulate the drug release rate and improve bioavailability.

The advantages of nanoparticles are strong permeability, good stability and easy modification. Nanoparticles effectively improve the bioavailability of active substances. Although nanoparticles have a certain targeting capability in vivo, this targeting is passive and other measures should be taken to improve targeting. The surface modification of nanoparticles can prevent macrophage phagocytosis and change the distribution of the nanoparticles. The disadvantages in the application of SLNs, such as large volume, early release and easy gelation, must be overcome [57].

## 5. The Research Status of PhG Delivery Systems

The embedding of PhGs in the delivery systems can effectively improve its bioavailability. In recent years, many scholars have devoted themselves to studying the combination of PhGs and various delivery systems, determining a series of characterization of the delivery systems through in vitro experiments and studying its effect and mechanism in vivo experiments.

### 5.1. Echinacoside

Echinacoside (ECH) is a representative PhG in the plant kingdom and appears as a white crystalline powder. Many traditional Chinese anti-aging medicines contain ECH, such as *Cistanche deserticola*, Orobanche and *Rehmannia glutinosa*. The content of ECH in *Cistanche tubulosa* can be up to 30%. Recent studies have shown that the main pharmacological effects of ECH are antioxidation, nerve protection, liver protection and memory improvement. The results showed that the scavenging activity of ECH on DPPH was the strongest [58]. Currently, ECH still has some problems in vivo, such as intestinal malabsorption and poor bioavailability. For example, the bioavailability of ECH in rats following the oral administration of ECH (100 mg/kg) was only 0.83% [59].

Li et al. [10] prepared an ECH–phospholipid complex (PHY) using solvent evaporation. PHY was characterized by differential scanning calorimetry (DSC) and infrared spectroscopy (IR) and then its physicochemical properties, intestinal absorption and bioavailability were studied. DSC and IR images showed that ECH and the phospholipids formed weak bonds. The complex improved the lipophilicity of ECH compared with both ECH alone and an ECH phospholipid mixture. PHY significantly increased the absorption and bioavailability of ECH in rats. Su et al. [60] used polylactic glycolic acid (PLGA) as the drug carrier to prepare ECH nanoparticles using the multiple emulsion solvent evaporation method. In the early stages, single factor experiments were carried out to investigate the concentrations of PLGA and poloxamer 188 (F68) and the volume ratio of internal aqueous phase to oil phase. The results showed that when PLGA was 5%, the corresponding F68 and volume ratio of internal water phase to oil phase were 2.57% and 0.86, respectively, and that the particle size and entrapment efficiency of the ECH nanoparticles were close to the theoretical value. Transmission electron microscopy observation showed that the nanoparticles were round and had a smooth surface and a uniform size and distribution. In vitro release tests showed that the ECH nanoparticles had a sustained release effect. Chen et al. [61] explored the effects of different Supplementary materials on ECH solid lipid nanoparticles and selected the emulsion curing method to prepare SLNs. The zeta potential was used to predict the stability of nanoparticles and ultrafiltration was used to determine the encapsulation efficiency. Glycerin monostearate was used as the lipid material and lecithin and myrj52 were used as surfactants. The results showed that with increasing myrj52 concentrations, the particle size of the nanoparticles decreased, the encapsulation efficiency increased and the zeta potential increased. With the increase in glyceryl monostearate concentration, the particle size increased significantly, the encapsulation efficiency decreased slightly and the zeta potential decreased. Increasing the lecithin concentration increased the particle size significantly and decreased the zeta potential and the encapsulation efficiency. Xue et al. [62] characterized the optimized SLNs and evaluated them using eye cells in vitro. The results showed that ECH existed in a molecular dispersion state in SLNs and that ECH could be delivered into eye cells via SLNs. This study verified the effectiveness of SLNs as an eye drug delivery system.

### 5.2. Acteoside

Acteoside (ACT), which is also known as verbascoside, ergosterol and cimicifugin, is widely distributed in the plant kingdom and is commonly found in *Rehmannia glutinosa*, *Cistanche deserticola* and *Osmanthus fragrans*. ACT is a white or light yellow crystalline powder. Its structure is characterized by a central glucose unit connected with the α-hydroxyl group of phenylethanol at the C_1_ position, rhamnose at the C_3_ position and amphetamine at the C_4_ position [63]. It has anti-tumor, anti-inflammatory and neuroprotective effects. The content of ACT in medicinal plants is low and its stability is poor. The structural characteristics of its multi-hydroxyl group determine its ease of oxidation, which greatly limits the application of ACT.

Isacchi et al. [64] used a thin layer evaporation technique to prepare ACT liposomes. The raw materials were Phospholipon 90G (P90G), cholesterol and ACT. The average particle size of the obtained liposomes was about 120 nm, the encapsulation efficiency was 30% and the amount of ACT released was 82.28 ± 1.79%. Liposomes could improve the stability of ACT by inhibiting its hydrolysis. The performance of the liposomal formulation was compared with that of the free drug using the paw pressure test in chronic constriction injury rats. The results showed that the liposomal formulation showed a longer lasting antihyperalgesic effect in comparison with ACT saline solution. Kalantari et al. [65] prepared different self-emulsifying combinations using the water and oil dilution method to study the protective effect of the self-nano emulsifying drug delivery system (SNEDDS) on a Plantago lanceolata leaves extract containing ACT. The results showed that SNEDDS compositions potentiated the free radical scavenging activity of the lanceolata extract compared with that of the non-encapsulated lanceolata extract. Cytotoxicity tests, dissolution tests and other experiments showed that the lanceolata extract SNEDDS could be used to deliver active natural compounds in a stable, efficient and safe manner. Zhou et al. [66] prepared liposomes through an ethanol injection method. ACT was encapsulated within liposomes with chitosan. The results showed that the ACT liposomes (ACT-Lip) were spherical and had an average diameter, zeta potential, encapsulation efficiency and relative bioavailability of 78.49 ± 1.44 nm, −4.93 ± 0.79 mV, 81.06 ± 3.48% and 217.62%, respectively. After encapsulation in chitosan modified liposomes (Cs-Lip), the characterization parameters were improved. In contrast to ACT-Lip, ACT–CS-Lip showed a reduced in vivo release rate and enhanced storage stability. Xue et al. [67] synthesized chitosan polyethylene glycol-poly lactic acid (mPEG-PLA) nanoparticles and embedded ACT, plasmid DNA (pDNA) and nerve growth factor (NGF) to construct a nanomicelle composite (APPDN). The results showed that the size of the APPDN nanoparticles was about 160 nm. In addition, they had a positive charge and low toxicity and were biodegradable. APPDNs had significant neuroprotective effects in N-methyl-4-phenyl-1, 2, 3, 6-tetrahydropyridine (MPTP)- induced mouse models of Parkinson’s disease.

### 5.3. Salidroside

The molecular formula of salidroside (SAL) is C_14_H_20_O_7_ and its relative molecular weight is 300 Da. It has a glycoside chemical structure. SAL is colorless with a transparent needle-like crystal. It is soluble in water, ethanol and n-butanol and is slightly soluble in acetone and ether. Its aglycone tyrosol structure is p-hydroxyphenylethanol, with a molecular formula C_8_H_10_O_2_ and a relative molecular weight of 138 Da. SAL can be hydrolyzed into one molecule of glucose and one molecule of aglycone under the action of acids or enzymes. It can protect the nervous system, has anti-fatigue effects and strengthens the heart. Pharmacological studies have shown that the absorption of SAL in the gastrointestinal tract is poor and its bioavailability in vivo is low [68].

Chen [69] studied the preparation of compound gum (BCC) by modifying bovine serum albumin (BSA) with orange peel pectin (CPP), which is a waste recycling resource. Through the characterization of certain properties, the preparation mechanism was analyzed and SAL and vitamin C (VC), with poor stability, were selected for encapsulation and simulated release. The entrapment efficiency of SAL in the functional food was about 50%. The simulated in vitro release of SAL showed a gastrointestinal sustained-release function and had certain pH and ionic stability, which indicated that BCC has a potential application value for the delivery of SAL and other functional foods. Li [70] used chitosan and sodium alginate as the wall of the microcapsules. SAL drug particles formed the core of the microcapsule and the electrostatic attraction layer by layer nano self-assembly (LBL) technology was used to microencapsulate SAL in chitosan and sodium alginate to prepare the sustained-release microcapsules. The encapsulation efficiency of the microcapsules was higher than 55% and the time required for the microcapsules to release more than 80% of the active substance was less than 8 h with an increase in the number of wall layers, which indicated that the polyelectrolyte encapsulated microcapsules could slow down the release rate of SAL. Yang et al. [71] prepared a SAL microemulsion using the hydrophilic lipophilic balance (HLB) method. The average diameter was 65.56 ± 4.62 nm and the microemulsion was stable at room temperature. The results of in vitro transdermal experiments showed that the SAL microemulsion had good stability and a high transdermal transfer rate, which significantly improved the permeability of SAL. Zhang et al. [72,73] used Span 40, cholesterol and sodium dodecyl sulfate (SDS) as stabilizers to prepare lipid vesicles for SAL transdermal delivery. At concentrations of 0.05–0.40% (w/*v*), SDS significantly improved the stability of the nanocapsules. The transdermal flux and skin deposition of SAL were the highest when the molar ratio of Span 40 to cholesterol was 4:3. Xia et al. [74] prepared SAL nanoliposomes (SNLs) using the ethanol injection method. X-rays were orally administered to mice following whole body irradiation. The results showed that the SNLs had a stronger radiation protection ability than SAL solution and that they protected the blood system of the mice, inhibited peripheral blood lymphocyte apoptosis to a great extent, decreased the level of lipid peroxidation in liver tissue and enhanced the antioxidant capacity of the system.

Table 3 lists the application of other kinds of PhGs in different delivery systems, and introduces the preparation methods and physical and chemical characterizations. Although studies have proved that a variety of delivery systems can effectively improve the bioavailability of PhGs, there are few studies on the in vivo safety of delivery systems. Further research on the absorption mechanism of various types of PhG delivery systems through the small intestine and an evaluation of their in vivo efficacy needs to be carried out.

## 6. Safety Evaluation and Existing Problems

With the deepening of research, the application of PhGs and their delivery systems within the field of food is also expanding; and safety has become the focus of attention. The acute toxicity test showed that no mice died within 14 days after continuous gavage and that the average body weight increased by more than 50%. The maximum tolerance of mice to the extraction of *Cistanche deserticola* PhGs was 119.46 g/kg and the dose was 882 times that used for adults, indicating that the toxicity of *Cistanche deserticola* PhGs was very low, the clinical application was safe and reliable and it had benefitted from the further research and development [80]. Other experiments showed that the repeated administration of *Lamiophlomis rotata* PhGs extract (L-PhGs) to mice for 30 days showed no obvious toxicity, indicating that L-PhGs were safe for long-term use [81]. After the long-term administration of *Monochasma savatieri* PhGs extract for 90 days, rats had a certain degree of damage to liver and lung [82]. The acute toxicity test showed that within the dose range of 100 mg/mL, the PhG nanoemulsion had no toxicity and intranasal administration had no significant effect on nasal mucosa or important organs, indicating that the PhG nanoemulsion had a good level of safety [83].

In vitro experiments showed that the five PhGs in *Chirita eburnea* had weak cytotoxic side effects on SPC-A1 cell lines and no cytotoxicity on HepG2, SGC7901 or other cell lines [84]. Acteoside liposomes can be maintained for 6 months at 4 ± 1 °C. The stability study showed that the liposomes were physically stable within 90 days. During this period, ACT did not leak and the size of the liposomes did not change. After 3 months, ACT remained at 91% in the liposomes and decreased to 80% after 6 months, but the ACT did not degrade throughout the whole process [64].

There are still some problems associated with delivery systems: (1) different standards and models are used to evaluate the physical and chemical properties and bioavailability of delivery systems, and there is a lack of pertinence and contrast; (2) some active substances are inhibitors or promoters for transporters and metabolic enzymes and a change in bioavailability would also alter these functions; and (3) the targeted transport of PhGs in delivery systems cannot achieve accurate positioning, which leads to a decrease in the absorption and bioavailability of PhGs.

## 7. Conclusions

PhGs have a wide range of chemical, biological and physiological activities. However, their limited stability and poor bioavailability must be solved in order to make these components better able to meet people’s needs in terms of nutrition and health. In this paper, the main biological activities, digestion, absorption and metabolism of PhGs and the application status of three typical PhGs and their delivery systems were reviewed. The various research results reported in this paper show that the delivery system has a significant protective effect on the PhGs under various conditions in vivo and in vitro and effectively enhances the stability and bioavailability of PhGs. Furthermore, the delivery system can realize the targeted transport and release of PhGs, so that they can provide a specific therapeutic effect.

However, there are still a series of problems, such as different evaluation standards and unclear safety in vivo. Therefore, the factors affecting the bioavailability, absorption, metabolism, transport process and mechanisms of PhGs still need to be explored further so as to improve the safety of PhG delivery systems and build a high-performance delivery system.

## Figures and Tables

**Figure 1 foods-11-00769-f001:**
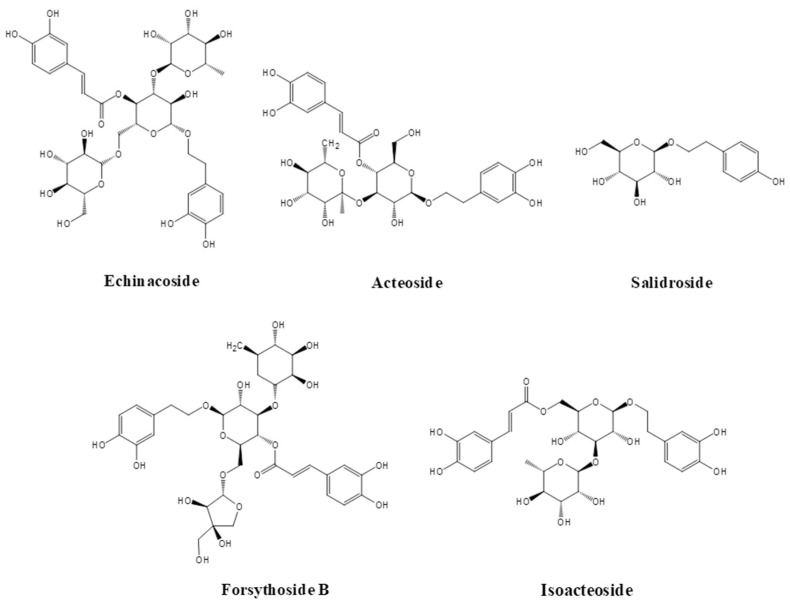
The molecular structures of several typical PhGs.

**Figure 2 foods-11-00769-f002:**
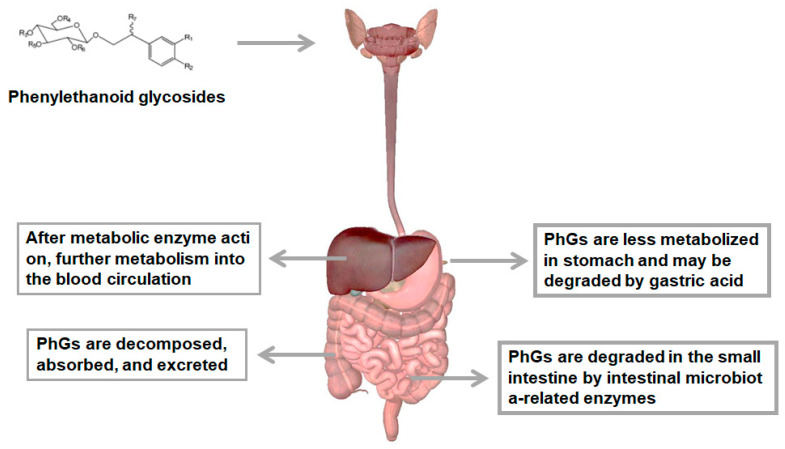
The absorption and metabolism of PhGs in vivo.

**Figure 3 foods-11-00769-f003:**
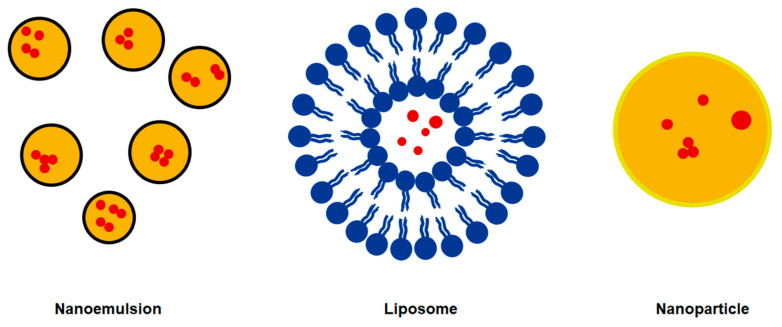
Three typical delivery systems.

**Table 1 foods-11-00769-t001:** Research status of PhG bioactivity.

Bioactivity	PhG	Disease Model	Mechanism of Action	References
Anti-inflammatory	Acteoside	Alcohol-induced HepG2 cells	the inhibition of cytokine production and downregulation of NFκβ/Iκβ signaling was observed	[26]
	Echinacoside	Dextran sulfate sodium (DSS)-induced colitis in mice	The expression of growth factor TGF-B1 in cultured intestinal epithelial cells was stimulated	[27]
Anti-aging	Echinacoside	β-amyloid (Aβ)-induced toxicity of *Caenorhabditis elegans*	ECH triggered DAF-16 nuclear localization and upregulated two of its downstream targets: sod-3 and hsp-16.2	[28]
	Acteoside, AE	SAMP8 mice	an increase in the number of helper lymphocytes and the regulation of the dynamic balance of Th1 and Th2 immune inflammatory cytokines was found	[29]
Anticancer	Salidroside	The human breast cancer cell line MCF-7	the downregulation of the expression of MMP-2 and MMP-9 was seen, thereby inhibiting cancer cell invasion and metastasis	[30]
	Salidroside	Bladder cancer cells	Through the autophagy/PI3K/Akt and MMP-9 signaling pathway, viability was reduced	[31]
Analgesic	PhGs of *C. deserticola* (component: Echinacosid, acteoside and tubuloside B)	Glutamate and capsaicin models (injected into the plantar region of mice)	Could be mediated through ionotropic metabotropic receptors, which regulate the activity of TRPV1	[32]
Immunomodulatory	PhGs of *C. deserticola*	Dendritic cells (DCs) of mice	Enhanced the expression level of CD86 and MHCII on the surface of CD11c++DCs	[33]

**Table 2 foods-11-00769-t002:** The pharmacokinetic parameters of different PhGs.

Compound	Dose (mg/kg)	Model	C_max_ (µg/mL)	T_max_ (h)	T_1/2_ (h)	AUC_0__→__t_ (ng.h/mL)	AUC_0__→__∞_ (ng.h/mL)	MRT_0__>→__t_ (h)	Ref.
Acteoside	20 ig	Rats	0.16 ± 0.05	0.17 ± 0.08	1.49 ± 0.28	164.98 ± 21.24	181.13 ± 20.55	1.41 ± 0.13	[43]
Acteoside	10 ig	Dogs	0.42 ± 0.10	–	1.48 ± 0.16	788.0 ± 145.7	802.8 ± 147.8	1.93 ± 0.15	[44]
Acteoside	8 ig	Rats	0.19 ± 0.05	0.25 ± 0.12	2.51 ± 1.00	544.8 ± 127.2	566.8 ± 127.3	–	[45]
Echinacoside	20 ig	Rats	0.37 ± 0.05	0.27 ± 0.08	1.23 ± 0.33	–	2759.3 ± 79.46	5.76 ± 2.59	[46]
Salidroside	75 iv	Dogs	96.16 ± 8.59	0.25	2.006 ± 0.615	180.3 × 10^3^ ± 30.6 × 10^3^	189.3 × 10^3^ ± 32.1 × 10^3^	2.275 ± 0.704	[47]
Salidroside	20 ig	Rats	4.50 ± 1.48	0.3 ± 0.1	1.0 ± 0.2	8.28 × 10^3^ ± 1.61 × 10^3^	8.32 × 10^3^ ± 1.62 × 10^3^	1.4 ± 0.2	[48]
Forsythiaside	10 iv	Rats	–	–	1.21 ± 0.33	7687 ± 403.7	–	0.91 ± 0.77	[49]
Isoacteoside	10 ig	Rats	0.06 ± 0.02	0.4 ± 0.2	4.6 ± 3.1	70.9 ± 26.9	87.0 ± 40.0	–	[50]
Martynoside			0.004 ± 0.001	3.1 ± 3.6	9.0 ± 2.7	23.6 ± 6.9	39.5 ± 15.5		
Crenatoside			0.01 ± 0.001	6.8 ± 1.1	3.4 ± 3.1	64.7 ± 14.5	76.0 ± 30.0		

Abbreviations: AUC_0__→__∞_, area under the concentration–time curve extrapolated from zero up to infinity; AUC_0__→__t_, area under the concentration–time curve calculated from zero up to the last measured concentration; C_max_, maximum concentration; F, bioavailability; ig, intragastric gavage; iv, intravenous injection; MRT, mean residence time; T_1/2_, elimination half-life; T_max_, time maximum concentration.

**Table 3 foods-11-00769-t003:** The application of PhG delivery systems.

PhG	Delivery System	Preparation Method	Characteristic Parameters or Effect	References
Herba cistanches PhGs	Liposomes	Film dispersion method, two encapsulation film dispersion method and reverse evaporation method	Particle size (212.70 ± 1.27) nm, zeta potential (−37.85 ± 0.68) mV and encapsulation rate (38.46 ± 7.85)%	[75]
Herba cistanches PhGs	Liposomes	Vacuum freeze drying method	Particle size (207.7 ± 2.31) nm, zeta potential (−61.5 ± 3.18) mV, encapsulation rate (37.2 ± 0.30)% and drug loading (3.77 ± 0.07)%	[76]
Herba cistanches PhGs	Liposomes	Two encapsulation film dispersion method	Particle size 212.7 nm, zeta potential 35~50 mV, encapsulation rate (38.46 ± 7.85)% and inhibition of rat hepatic stellate cell (HSC) proliferation	[77]
Salidroside	Chitosan microspheres	Emulsion crosslinking method	Particle size 0.56~5.01 μm, encapsulation rate > 77.58% and drug loading > 23.29%	[78]
Salidroside	PLGA nanoparticles	The double emulsion (W/O/W) method	Particle size (275.3 ± 44.0) nm, polydispersity index (0.302 ± 0.102), zeta potential (−6.98 ± 2.99) mV and encapsulation rate (32.63% ± 0.73)%	[79]

## Data Availability

Data is contained within the article.

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
