# Peer review of "Research Progress and Trends of Phenylethanoid Glycoside Delivery Systems"

_foods, 2022, doi:10.3390/foods11050769_

Round 1
Reviewer 1 Report
The manuscript Research Progress and Trends of Phenylethanoid Glycoside Delivery Systems describes an interesting topic on a group of phenolic compounds with interesting pharmacological activities. However, the article should be somewhat shortened so that it wold enable better focus on the topic. Additional errors that should be fixed before publication are as follows:
1.
Fig. 1. The quality of figure is not satisfactory.
2.
References should be written in accordance with the instructions for authors. They must be numbered in order of appearance in the text (including table captions and figure legends) and listed individually at the end of the manuscript.
3.
line 51 "For example, Forsythia A showed good resistance" is not clear. What is Forsythia A? A plant whose extract was used? Please be more exact in the terninology.
4.
The plant names mentioned in the manuscript should be written using binomial names, in italic. The species author's name and the plant family should be mentioned on the first use,
5.
line 55: "hypoxic-Ischemic" change to "hypoxic-ischemic"
6.
line 58: "neuroprotective effect against PC12" Why would there be effect AGAINST cells? please rephrase.
7.
Table 1. Please use horisontal lines between different activities. The way data is presented, it is not entirely clear which PhG presents which activity.
8.
Sections 4.1.-4.3. These sections should be combined into one and shortened because they just repeat well-known information, and are not directly related to the subject.
9.
Conclusion should be reduced to about 1/3 or 1/2 of current size. Furthermore, it should be completely re-written because in its current form it does not offer the conclusion on the presented research. Thus, a shortened version should be provided that offers summarized conclusion of the information presented in the previous sections.
Reviewer 2 Report
The manuscript attempts to review the biological activities and the delivery systems of a class of glycosides (phenylethanoid glycosides) coming from natural sources as well as the embedding technologies used for some main compounds (echinoside, acteoside, and salidroside). Since all reviews are based on many references, references should be numbered in order of appearance in the text, according to instructions of authors as suggested by Foods journal. Therefore, the whole manuscript should be revised in this context.
The review should be reorganized in a better way, more specifically, adding all the information in Tables and discussing the findings in detail and in comparison. Tables 1 and 2 are not discussed in the main manuscript. Authors should incorporate and comment on the research studies of the Tables. Overall, my recommendation is major revision.
Round 2
Reviewer 2 Report
The manuscript has been sufficiently improved and revised, therefore I suggest the publication in its revised form.
